# Effect of Hydrophobic Treatments on Improving the Salt Frost Resistance of Concrete

**DOI:** 10.3390/ma13235361

**Published:** 2020-11-26

**Authors:** Guo Li, Chunhua Fan, Yajun Lv, Fujun Fan

**Affiliations:** 1Jiangsu Key Laboratory of Environmental Impact and Structural Safety in Engineering, China University of Mining and Technology, Xuzhou 221116, China; guoli@cumt.edu.cn (G.L.); ts18030126p31@cumt.edu.cn (C.F.); 2School of Architecture, North China University of Water Resources and Electric Power, Zhengzhou 450045, China; x20201140959@stu.ncwu.edu.cn

**Keywords:** hydrophobic treatment, concrete, salt scaling, contact angle, freeze-thaw cycle

## Abstract

Hydrophobic treatment is an important method to improve the waterproof properties of concrete. To evaluate the effectiveness of hydrophobic treatments on improving the salt frost resistance of concrete, two representative commercial ordinary water repellent agents of silane and organosilicone emulsion were selected, and concrete specimens with three water/cement ratios were fabricated. After the application of repellent agents on concrete surfaces, accelerated saline (5% MgCl_2_) freeze-thaw cycles were conducted on the specimens. The mass losses and relative dynamic modulus of elasticity (RDME) of concrete were tested periodically. The contact angles and water absorption ratios of concrete with and without hydrophobic treatments were also tested. Results showed that the repellent agents could substantially enhance the hydrophobicity of concrete and greatly reduce its water absorption. Different repellent agents exerted diverse improvements on concrete hydrophobicity. Meanwhile, the repellent agents could improve concrete resistance against salt scaling and RDME losses to a certain degree, and concrete with strong hydrophobicity showed relatively high salt frost resistance. However, the ordinary water repellent agents cannot achieve the same enhancement on salt frost resistance of concrete as that on the water hydrophobicity of concrete. With saline freezing and thawing cycles, the hydrophobic layer formed by the repellent agents on superficial concrete was destroyed gradually. As a result, the salt frost resistance of concrete from the hydrophobic treatments was ultimately lost.

## 1. Introduction

Salt frost damage has consistently been an important concrete durability issue in the northern region of the world [1,2,3]. Therefore, scholars and experts have implemented various measures, mainly including reducing the water/cement (w/c) ratio of concrete and incorporating mineral admixtures, nanomaterials, fiber materials, or air-entraining agents [4,5,6,7,8,9,10], to improve the salt frost resistance of concrete; among these measures, the most effective one for concretes with the same w/c ratio is the addition of an air-entraining agent. Concrete with a high air content generally has high frost resistance [4]. However, the improvement of the frost resistance of concrete by an air-entraining agent often leads to a reduction in concrete strength [11,12]. Hence, methods (except for air entraining) that can effectively enhance the salt frost resistance of concrete are desired. Hydrophobic treatment of concrete surfaces, which is one of these methods, is a simple and promising approach.

Hydrophobic treatment can line concrete surface pores and lower concrete surface energy, then change concrete from being hydrophilic to being hydrophobic to restrain the invasion of water and corrosive media into concrete. Many researchers, including Dai et al. [13], Zhao et al. [14], Shen et al. [15], and Courard et al. [16], have attempted to apply some commercial repellent agents (such as organosilicone emulsion, silane, siloxane, or the mixture) on concrete to improve its water impermeability, chloride resistance, carbonation resistance, or steel bar’s corrosion risk, and encouraging results have been achieved. The durability problems of reinforced concrete structures are mostly related to water intrusion, and frost damage of concrete is not an exception. If moisture intrusion into concrete can be effectively restrained, then the frost resistance of concrete may be improved.

Several researchers have studied the frost resistance of concrete treated with hydrophobic agents, and satisfactory results have been obtained. Basheer and Cleland [17] reported that the presence of pore liners (silane and siloxane) improves the freeze-thaw durability of concrete in freezing media of fresh water and saltwater; fresh water is expected to produce internal damage, and saltwater is expected to produce surface scaling. Dang et al. [18] studied the salt-scaling performance of concrete treated with seven commercial products, namely, three concrete sealers, two crack sealants, and two water repellents of silane; they found that 90% or more of mass losses of concrete with all surface treatments are avoided after 15 freeze-thaw and wet-dry cycles. Zhao and Yu [19] conducted rapid freeze-thaw cycle experiments on silane-impregnated concrete and discovered that hydrophobic treatments on the concrete surface remarkably reduce the water absorption of concrete and greatly improve concrete frost resistance. On the basis of the results of capillary suction of deicing solution and freeze-thaw testing (CDF) after 56 freeze-thaw cycles, Li and Ba [20] concluded that hydrophobic treatment by one homemade organosilicone repellent reduces the mass loss ratio of concrete by 99% and the loss of relative dynamic modulus of elasticity (RDME) by 49%.

The aforementioned studies have revealed that some hydrophobic treatments can substantially mitigate the frost damage of concrete, which gives people an impression that a simple hydrophobic treatment can significantly improve the frost resistance of concrete. However, some findings about the effectiveness of hydrophobic treatments in the salt scaling performance of concrete differ [21,22,23]. On the basis of the results of CDF experiments on pavement concrete, Frentzel-Schirmacher [21] determined that in the first 28 freeze-thaw cycles, the salt scaling amount of silane-impregnated concrete is much lower than that of untreated concrete. However, the salt scaling amount of silane-impregnated concrete increases afterward, and it ultimately exceeds that of the compared concrete after 40 cycles. Liu and Hansen [22] reported that when silane is applied, the surface scaling of concrete is substantially reduced only in the early stage, and the improvement effects on the internal frost damage of concrete are limited. Guo et al. [23] tested the performances of six types of coatings in freeze-thaw condition with 7% NaCl, including polymer coatings, polymer modified cementitious coatings, and a silicone hydrophobic agent. They found that the polymer-modified cementitious coatings demonstrated the best performance, while the silicone hydrophobic agent could only provide a slight improvement.

To date, only a few studies have been conducted on improving the salt frost resistance of concrete by hydrophobic treatment, and several disagreements remain regarding the efficacy of hydrophobic treatment in enhancing the salt scaling performance of concrete. Whether a simple hydrophobic treatment on concrete surface can afford durable and effective protection on the salt frost resistance of concrete requires further research [24,25]. In the current study, two representative commercial repellent agents were applied to concrete. The relationships among concrete contact angles, water absorption, and mass and RDME losses caused by salt frost damage were studied to evaluate the efficacy of different repellent agents in improving the salt frost resistance of concrete, and the mechanism of repellent agents on improving the salt frost resistance of concrete was analyzed. This study can provide a scientific guide for the application of hydrophobic treatment in concrete structures in cold regions.

## 2. Experimental

### 2.1. Raw Materials and Specimen Fabrication

According to China’s specification JGJ 55-2011 [26], three types of concrete with w/c ratios of 0.4, 0.5, and 0.6 that can represent the concrete types in the current civil engineering projects were designed as the reference concrete. Their mixture proportion is shown in Table 1. Among the concrete raw materials, the cement was P·O 42.5 ordinary Portland cement according to China’s specification GB 175-2007 [27], the fine aggregate was natural river sand with a fineness modulus of 2.47, the coarse aggregate was crushed limestone with 5–20 mm continuous particle size, the mixing water was ordinary tap water, and the water reducer was a polycarboxylate water reducer. Two representative commercial repellent agents were adopted. One was BS4004 silane emulsion (called BS4004 hereinafter) produced by Wacker Chemicals Co. Ltd. (Munich, Germany) and the other was C3033 organosilicone emulsion (called C3033 hereinafter) produced by Wuhan Zhiruida New Material Co.; Ltd. (Wuhan, China).

Specimens with two sizes of 100 × 100 × 100 mm^3^ and 100 × 100 × 400 mm^3^ were used for water absorption and salt frost experiments, respectively. The concrete was mixed with a forced mixer, and specimens were compacted using a plate vibrator. After demolding at the age of 24 h, the specimens were placed in a water tank with a water temperature of 20 °C for curing until the age of 28 days. Before the application of repellent agents, the specimens were placed in an oven (60 °C) to dry for 48 h. Following the instructions recommended by the vendors, the two repellent agents were brushed twice on the surface of the specimens with a time interval of 2 h to ensure hydrophobic effects. Typical specimens applied with a repellent agent are shown in Figure 1. The following experiments began after drying in an indoor natural environment for 7 days. Three specimens of the same type in one test were fabricated, and the mean value of the three data was regarded as a representative value.

### 2.2. Experimental Methods

Concrete contact angle and water absorption are important indexes to reflect the hydrophobicity of concrete. The contact angles in this study were obtained in accordance with the shape of a water droplet (5 μL) on a concrete surface using a USB digital microscope [28]. The four sides of cubic specimens were sealed with paraffin wax to obtain the water absorption ratio of concrete, and only one bottom side was kept for hydrophobic treatment. Then, the specimens were soaked in a water tank with a depth of 10 mm (Figure 2) and weighed regularly to calculate the water absorption ratio of concrete (Equation (1)).
(1)$$w_{c}=\frac{m_{t}-m_{0}}{m_{0}}\times 100\%,$$
where *w_c_* is the mass water absorption ratio of concrete (%) and *m*_0_ and *m_t_* are the initial masses of a concrete specimen and its mass after soaking for *t* time (g), respectively.

The salt frost tests on concrete were conducted in accordance with a rapid freeze-thaw cycle regime following the GB/T 50082-2009 standard [29], and as a common deicing salt in current road engineering, MgCl_2_ with 5% concentration (by mass) was selected as the corrosion medium [7,30]. The device adopted for the concrete salt frost experiments was a TR-TDRF-1 rapid freeze-thaw cycle machine produced by Shanghai Torrent Instrument Co. LTD (Shanghai, China). The specimens were submerged in the MgCl_2_ solution for 4 days in advance to achieve saturation before salt frost testing. Each freeze-thaw cycle lasted for approximately 4 h, and the thawing time was not shorter than 1 h. The central temperature of a specimen was controlled between −17 and +8 °C during the freezing and thawing processes. The appearance of the concrete specimens was regularly observed after every 25 cycles, and their masses and ultrasonic velocities were tested to calculate the mass loss ratio (Equation (2)) and RDME (Equation (3)) [25,30]. According to the specification in GB/T 50082-2009 [29], salt frost testing can be considered finished when the mass loss ratio of a concrete specimen reaches 5% or its RDME decreases to 60%.
(2)$$\alpha =\frac{m_{t0}-m_{t0}^{\prime }}{m_{t0}}\times 100\%$$
(3)$$\beta =\frac{E_{t}}{E_{0}}=\frac{v_{t}^{2}}{v_{0}^{2}}\times 100\%,$$
where *α* is the mass loss ratio of concrete (%); *m_t_*_0_ and *m_t_*_0_′ are the initial mass of specimens and their mass after salt frost (g), respectively; *β* is the RDME of concrete (%); *E*_0_ and *E_t_* are the dynamic modulus of elasticity of specimens before and after salt frost, respectively; and *v*_0_ and *v_t_* are the ultrasonic velocity of specimens before and after salt frost (m/s).

## 3. Results and Discussion

### 3.1. Contact Angles of Concrete with Hydrophobic Treatments

Water contact angle (*θ*) is a simple parameter to characterize the hydrophobicity of concrete. Generally, materials with a water contact angle of less than 90° are defined as hydrophilic, while materials with a water contact angle of more than 90° are defined as hydrophobic. Moreover, the lower water contact angle infers the more hydrophilicity of the material. Conversely, the higher water contact angle means the stronger hydrophobicity of the material [28,31]. The photographs of water droplets on the concrete surface with different water-repellent agents are shown in Figure 3. The shapes of the water droplets on the concrete surface changed markedly after the hydrophobic treatments. Concrete is a hydrophilic material; thus, the shape of a water droplet on the reference concrete was flat (Figure 3a). After the application of repellent agents BS4004 and C3033, the shapes of the water droplets became round (Figure 3b,c), indicating that the repellent agents substantially enhanced the hydrophobicity of concrete [18,24]. From the water droplets shown in Figure 3, the contact angles of untreated concrete, concrete with BS4004, and concrete with C3033 were determined to be 38°, 125°, and 102.7°, respectively. BS4004 and C3033 increased the contact angles of concrete by 2.3 and 1.7 times relative to that of the untreated concrete, respectively, and changed the concrete from being hydrophilic (*θ* < 90°) to being hydrophobic (*θ* > 90°).

### 3.2. Water Absorption of Concrete with Hydrophobic Treatments

The water absorption of concrete is also an important index of hydrophobicity [19,28,32]. The changes in the water absorption of concrete treated with different repellent agents are presented in Figure 4.

Concrete is rich in capillary pores and has strong hydrophilicity [33,34]. The water absorption ratios of the three types of concrete increased continuously with soaking time. The increment rates of the water absorption of concrete with different w/c ratios and water-repellent agents differed. The water absorption ratio of concrete with a high w/c ratio generally increased rapidly. The water absorption ratios of concrete increased slowly after hydrophobic treatments. Specifically, the water absorption ratio of the concrete treated with BS4004 was apparently lower than that of the concrete treated with C3033. These results are consistent with those in references [14,18,22].

With the extension of soaking time, the water absorption ratios of each concrete specimen stabilized due to the saturation of capillary water [32]. The water absorption ratios of concrete after hydrophobic treatments were lower than that of untreated concrete, and the water absorption ratio of the concrete treated with BS4004 was lower than that of the concrete treated with C3033. At a soaking time of 156 h, the water absorption ratios of the concrete (w/c = 0.6) treated with BS4004 and C3033 and the untreated one were 0.10%, 0.24%, and 0.99%, respectively. The water absorption ratios of concrete decreased by 89.9% and 75.7% when BS4004 and C3033 were used, respectively. This result confirms that the application of water-repellent agents on concrete is effective in reducing water absorption [13,17,19,35]. However, it is worth noting that hydrophobic treatments on concrete can only reduce the water uptake amount but cannot prevent exterior water from entering concrete completely.

### 3.3. Macromorphology of Concrete after Salt Frost Damage

With the development of the freeze-thaw cycle, typical salt scaling phenomena of concrete, such as mortar peeling and coarse aggregate exposure, gradually occur under the combined action of frost damage and salt attack [4,5,6]. Photographs of typical concrete specimens (w/c = 0.5) at different freeze-thaw cycles were obtained. They are shown in Figure 5.

As indicated in Figure 5, the hydrophobic treatments did not alter the appearance of the concrete surface; hence, no obvious difference was observed in the appearance of the specimens before and after hydrophobic treatments prior to the freeze-thaw cycles (Figure 5a,d,g). Valenza and Scherer [6] and Wu et al. [7] reported that when concrete is subjected to repetitive saline freeze-thaw cycles, a large mismatch occurs in the thermal expansion of ice and paste, which could create high tension during cooling. In the presence of deicing salt, the unfrozen liquid creates brine pockets that weaken the ice and promote the deterioration of concrete. When the swelling pressure surpasses the tensile strength of the concrete, the crack network extends outward, resulting in the cracking of the concrete surface.

With the development of the freeze-thaw cycles, the three types of concrete specimens in the current study exhibited evident salt scaling phenomena. Under the same freeze-thaw cycle, the damage extents of concrete with hydrophobic treatments were relatively lower than that of the untreated concrete (Figure 5c,f,i). The damage extent of the concrete treated with BS4004 was slightly lower than that of the concrete treated with C3033. This result confirms that the hydrophobic treatments alleviated the salt scaling damage on the concrete to a certain extent [17,18,19,20]. However, with continuous freeze-thaw cycles, the salt scaling phenomenon became severe even for the concrete specimens with hydrophobic treatments.

Typical pictures of the concrete specimens (w/c = 0.4) after the salt frost experiment are shown in Figure 6. All specimens demonstrated severe coarse aggregate exposure and losses of edges and corners, and the damage appearances of the specimens with and without hydrophobic treatments were similar [21,22].

### 3.4. Mass Losses of Concrete with Freeze-Thaw Cycles

The mass loss ratios of concrete treated with different repellent agents were plotted based on the measured masses of the concrete specimens during the saline freeze-thaw cycle. The plots are shown in Figure 7. In the initial period (approximately 25 cycles for concrete with w/c = 0.4), the mass loss ratios of all the specimens increased with the freeze-thaw cycles. The increment rates of the mass loss ratios of the specimens differed for concrete with diverse w/c ratios and repellent agents. Generally, the mass loss ratios of concrete with high w/c ratios increased rapidly, and the mass loss ratios of concrete with hydrophobic treatments and the same w/c ratio increased more slowly than that of the untreated concrete. This result suggests that the hydrophobic treatments were effective in alleviating the mass loss of concrete due to salt scaling, a result that agrees well with the findings in references [20,22].

Although the initial mass loss ratios of the concrete with repellent agents were generally lower than that of the untreated concrete, after reaching a certain number of freeze-thaw cycles, the mass loss ratios of the three types of concrete increased dramatically then reached the failure criterion of 5% mass loss ratio [29]. Therefore, the ultimate freeze-thaw cycle number of each concrete specimen can be determined using a linear interpolation method. The concrete with a w/c ratio of 0.4 was regarded as an example. The ultimate freeze-thaw cycles of the untreated, BS4004, and C3033 concrete were 264, 308, and 286, respectively. The service life improvements of the concrete with BS4004 and C3033 were only approximately 17% and 8.5%, respectively, and the average improvement was 12.7%. Similarly, the average improvements for the concrete with w/c ratios of 0.5 and 0.6 were 29.5% and 39.3%, respectively. Compared with the remarkable improvements in the contact angles and water absorption ratios of the concrete with hydrophobic treatments, the improvement in the mass loss ratio was unobvious.

### 3.5. RDME Losses of Concrete with Freeze-Thaw Cycles

The RDME of concrete is an index that reflects the development of concrete’s internal damage [4,5]. With the development of freeze-thaw cycles, water penetrates into concrete continuously. The repeated freezing (volume expansion) and thawing (volume reduction) of the capillary pore water in the concrete lead to cracking and damage accumulation in the concrete, and the RDME of concrete continues to decline. Figure 8 shows the development of the RDME of the concrete specimens with the number of saline freeze-thaw cycles.

The RDME of all specimens continuously decreased as the freeze-thaw cycles increased. Minimal differences were observed in the decrement rates of concrete with different w/c ratios and repellent agents. Generally, the RDME of the concrete with high w/c ratios decreased rapidly, and the RDME of the concrete after hydrophobic treatment became relatively slower than that of the untreated concrete. This result indicates that the hydrophobic treatments were beneficial to reduce the internal damage of concrete due to salt frost, but the overall improvements and the difference in the improvements of BS4004 and C3033 were relatively small. Such phenomenon indicates that although the water absorption ratios of concrete with hydrophobic treatments have been greatly reduced, the internal damages caused by saline freeze-thaw cycles have not been substantially reduced [22,23]. The reasons will be discussed in detail in the following Section 3.6.

The ultimate freeze-thaw cycles of concrete can be calculated in accordance with the failure criterion of 60% RDME [29,36]. The concrete with a w/c ratio of 0.4 was regarded as an example. The numbers of ultimate freeze-thaw cycles of the untreated concrete and concrete treated with BS4004 and C3033 were 212, 291, and 275, respectively. Compared with the ultimate freeze-thaw number of the untreated concrete, those of the concrete treated with BS4004 and C3033 were enhanced by approximately 37.2% and 29.5%, respectively. The average improvement was 33.3%. The ultimate freeze-thaw numbers of concrete based on RDME were smaller than those based on mass losses. This result is consistent with the finding of Penttala [5] stating that when the w/c ratio of concrete is below 0.42, internal deterioration, instead of surface scaling, is the governing freeze-thaw damage mechanism. Similarly, the average improvements in the number of ultimate freeze-thaw cycles of concrete with w/c ratios of 0.5 and 0.6 were 11.6% and 15.6%, respectively.

The mass loss and RDME of concrete were combined, and the ultimate freeze-thaw cycles of the concrete after hydrophobic treatments were ascertained (Table 2). Although the data showed certain discreteness, the development of the RDME and mass loss ratios of concrete was consistent. The average improvements of BS4004 and C3033 in the ultimate freeze-thaw cycles of concrete were 42% and 25%, respectively. These results indicate that hydrophobic treatments can only improve the salt frost resistance of concrete to a certain degree. In other words, the improvements in the salt frost resistance of concrete cannot be equal to the improvements in its hydrophobicity performance. Additionally, repellent agents with strong hydrophobicity can provide relatively high improvements. Hence, we deduce that with the emergence of superhydrophobic materials [37,38,39], further improvements of concrete salt frost resistance may be achieved.

### 3.6. Failure Mechanisms of Concrete with Hydrophobic Treatments

A schematic of the capillary water absorption of concrete after hydrophobic treatment is presented in Figure 9. The entry of water into concrete is a capillary phenomenon (Figure 9a). Hydrophobic treatments form a certain depth of hydrophobic layer through repellent agents impregnated into superficial concrete (Figure 9b), thus increasing the surface tension of this part of concrete capillary pores to restrain the penetration of moisture, Cl^−^, SO_4_^2−^, and other corrosive media [13,40]. However, the previous water absorption experiments indicated that this restraint of ordinary water repellent agents cannot completely prevent the entrance of external liquids, and a small amount of salt solution can still penetrate into concrete.

Zhang et al. [41] and Ma et al. [42] reported that continuous freeze-thaw cycles would generate new cracks in concrete, which would make concrete porous and greatly increase the capillary water absorption of ordinary concrete. A study by Liu and Hansen [22] indicated that the water absorption ratios of concrete with hydrophobic treatments increase substantially with the development of freeze-thaw cycles; these ratios can even approach or reach the level of water absorption ratio of untreated concrete. The penetration depth of ordinary water repellent agents in concrete is generally only a few millimeters [13,17,19]. Zeng et al. [43] reported that the thickness of the hydrophobic layer decreases with time because of the fracture of Si–O bonds caused by continuous exposure to alkaline pore solution. In the current study, with the development of the saline freeze-thaw cycle, the mortar on the concrete surface layer continued to peel off until the hydrophobic layer formed by repellent agents was completely destroyed; finally, the protection on concrete was completely lost (Figure 9c).

In summary, at the initial stage of salt frost on concrete, repellent agents can effectively restrain the entry of water into concrete and thus play a beneficial role in reducing the damage of concrete due to salt frost. However, with the development of saline freeze-thaw cycles, the formed hydrophobic layer is gradually destroyed. Thus, the salt frost resistance of concrete is gradually lost. The improvement effects of ordinary water repellent agents on the salt frost resistance of concrete and the way these repellents inhibit the entry of corrosive media (e.g., water and Cl^−^ and SO_4_^2−^ ions) cannot be maintained [44,45]. The efficacy declines with freeze-thaw cycles until it is completely lost. The improvement effects of ordinary water repellent agents on the salt frost resistance of concrete are only temporary and limited and should not be overestimated in engineering applications. Additionally, with the emergence of novel superhydrophobic materials, such superhydrophobic treatments can reduce the water absorption of concrete to a higher extent than ordinary hydrophobic treatments [36,37], which may produce satisfactory results on the salt frost resistance of concrete, and needs further studies in the future.

## 4. Conclusions

On the basis of experimental results and mechanism analysis of the salt frost performance of concrete with ordinary water repellent agents, some conclusions can be drawn:(1)Hydrophobic treatments can considerably enhance the hydrophobicity of the concrete superficial layer, convert concrete from being hydrophilic to being hydrophobic, and greatly reduce the water absorption of concrete. In this study, BS4004 and C3033 can increase the contact angles of concrete by 2.3 times and 1.7 times and decrease the water absorption ratios of concrete by 89.9% and 75.7%, respectively.(2)Ordinary water repellent agents can reduce the mass and RDME losses of concrete due to salt frost to a certain extent, then contribute to the improvement of salt frost resistance of concrete. In this study, the average ultimate freeze-thaw cycles of concrete can be enhanced 42% and 25% by BS4004 and C3033, respectively.(3)It is worth noting that the improvements on concrete hydrophobicity by ordinary water repellent agents are not equal to the improvements on concrete salt frost resistance. As the hydrophobic layer formed by repellent agents on superficial concrete is gradually destroyed during the saline freeze-thaw cycle, the improvement effects of hydrophobic treatments on the salt frost resistance of concrete are gradually lost.

## Figures and Tables

**Figure 1 materials-13-05361-f001:**
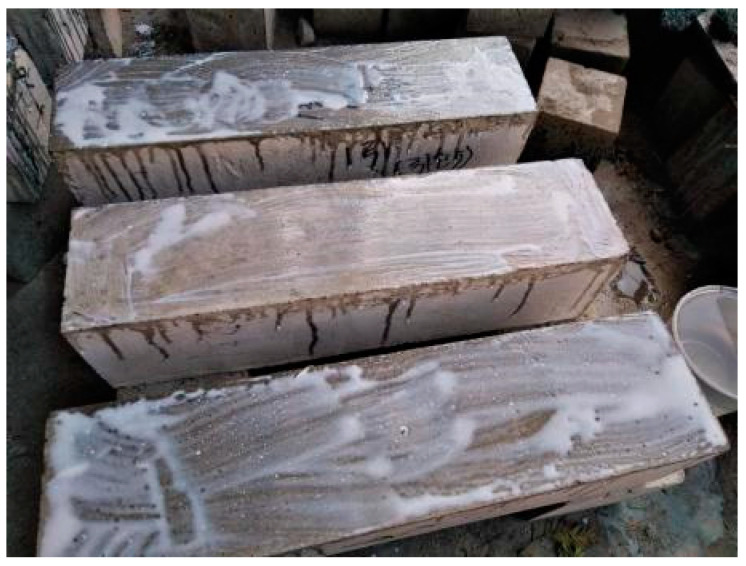
Prism concrete specimens applied with a repellent agent of BS4004.

**Figure 2 materials-13-05361-f002:**
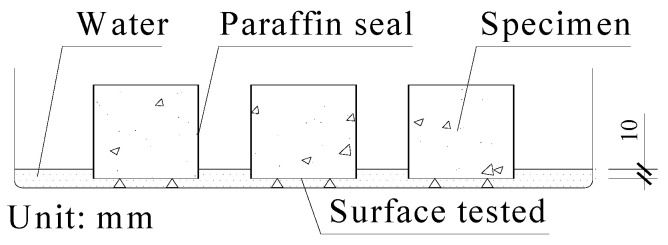
Schematic of a water absorption experiment for a single side of concrete.

**Figure 3 materials-13-05361-f003:**
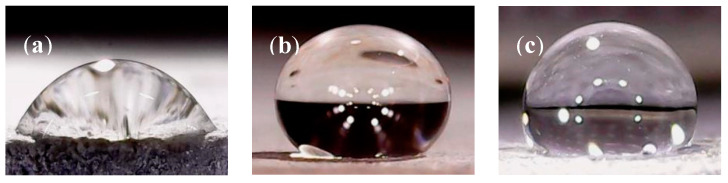
Water droplets on the surface of concrete treated with different repellent agents: (**a**) untreated, (**b**) BS4004, and (**c**) C3033.

**Figure 4 materials-13-05361-f004:**
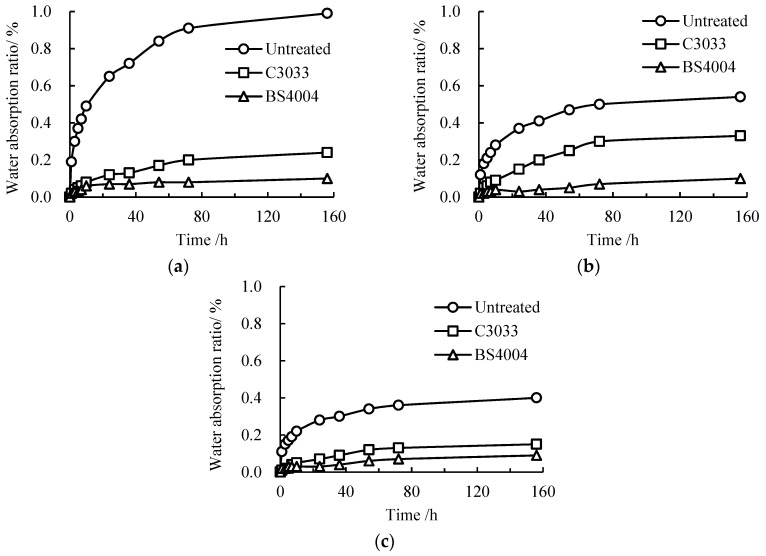
Development of the water absorption of surface-treated concrete with different w/c ratios: (**a**) w/c = 0.6, (**b**) w/c = 0.5, and (**c**) w/c = 0.4.

**Figure 5 materials-13-05361-f005:**
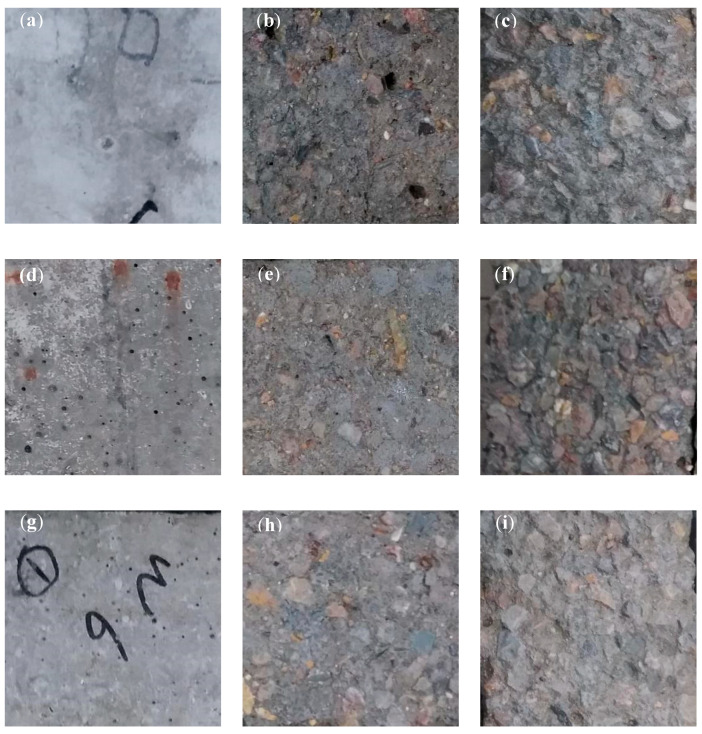
Typical appearance of the concrete specimens after different freezing/thawing cycles: (**a**) untreated (0 cycle), (**b**) untreated (50 cycles), (**c**) untreated (100 cycles), (**d**) BS4004 (0 cycle), (**e**) BS4004 (50 cycles), (**f**) BS4004 (100 cycles), (**g**) C3033 (0 cycle), (**h**) C3033 (50 cycles), and (**i**) C3033 (100 cycles).

**Figure 6 materials-13-05361-f006:**
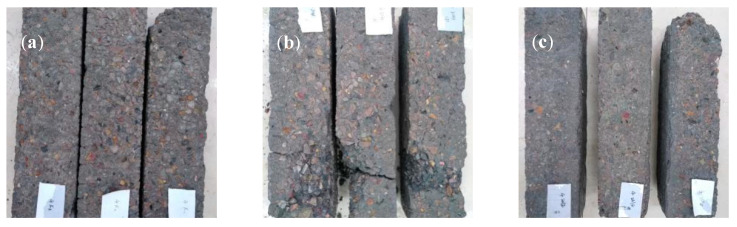
Typical concrete specimens after salt frost experiments: (**a**) untreated (275 cycles), (**b**) C3033 (275 cycles), and (**c**) BS4004 (300 cycles).

**Figure 7 materials-13-05361-f007:**
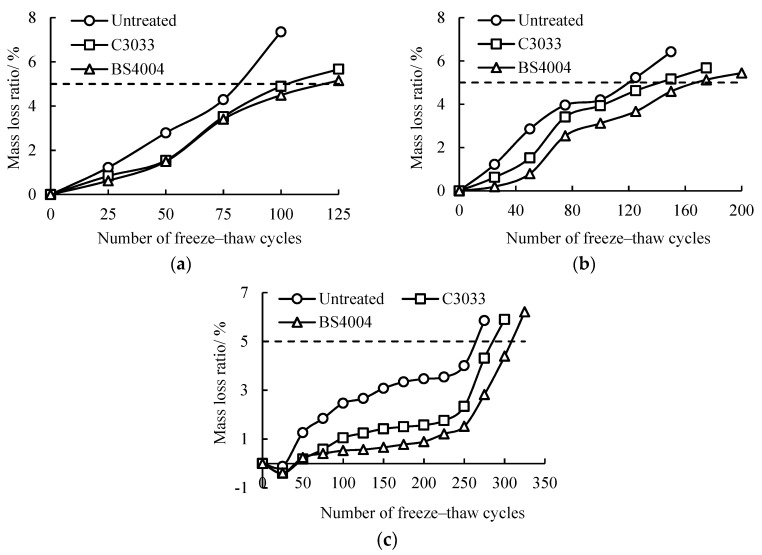
Mass loss ratios of concrete subjected to hydrophobic treatment with freeze-thaw cycles: (**a**) w/c = 0.6, (**b**) w/c = 0.5, and (**c**) w/c = 0.4.

**Figure 8 materials-13-05361-f008:**
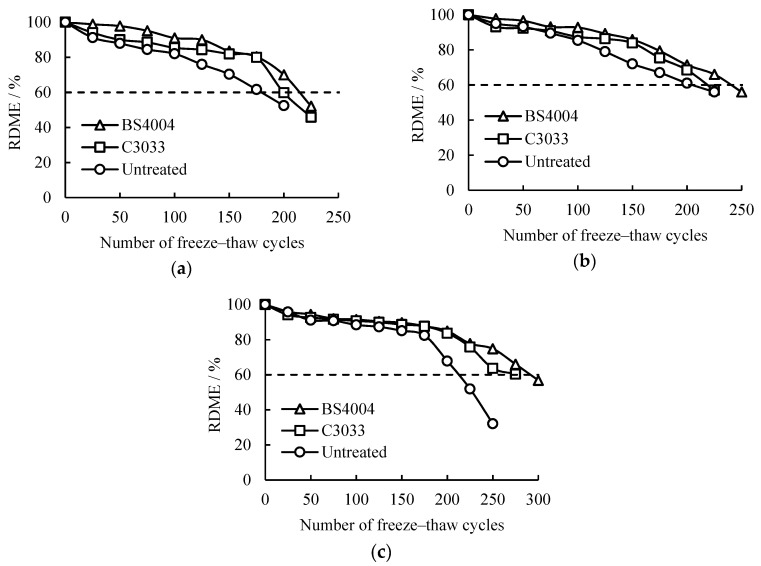
Relative dynamic modulus of elasticity (RDME) of concrete subjected to hydrophobic treatment with freeze-thaw cycles: (**a**) w/c = 0.6, (**b**) w/c = 0.5, and (**c**) w/c = 0.4.

**Figure 9 materials-13-05361-f009:**
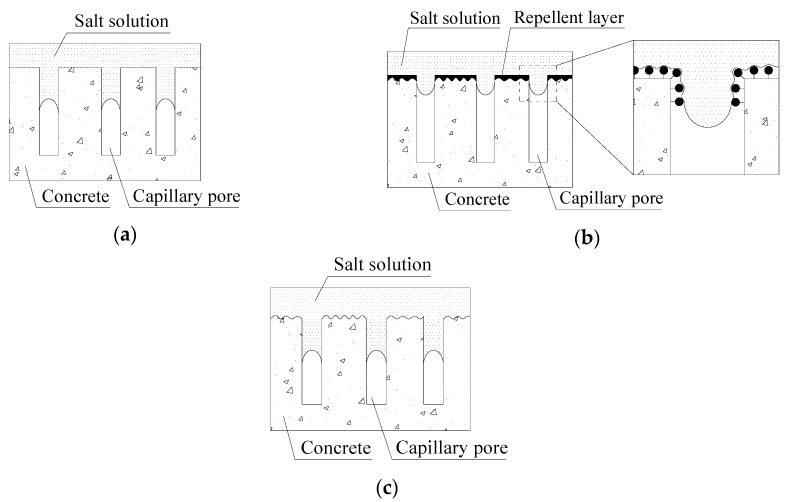
Schematic of the capillary water absorption of concrete after hydrophobic treatment: (**a**) untreated concrete, (**b**) concrete with hydrophobic treatment, and (**c**) concrete after salt frost.

**Table 1 materials-13-05361-t001:** Concrete mixture proportion/kg/m^3^.

w/c	Cement	Sand	Gravel	Water	Water Reducer
0.4	475	607	1178	190	2.4
0.5	360	687	1222	180	1.8
0.6	350	737	1153	210	1.7

**Table 2 materials-13-05361-t002:** Ultimate freeze-thaw cycles of concrete after hydrophobic treatments.

Item	w/c Ratio		
	0.6	0.5	0.4
Untreated	81	119	212
BS4004	119	169	291
C3033	103	142	275

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
