# Peer review of "Effect of Hydrophobic Treatments on Improving the Salt Frost Resistance of Concrete"

_materials, 2020, doi:10.3390/ma13235361_

Round 1
Reviewer 1 Report
The manuscript presents the research area of Effect of hydrophobic treatments on improving the salt frost resistance of concrete.
The research area covered by the manuscript can be of potential interest for the readers of this Journal of Materials.
The topic can be also considered topical, but the article has/presented limited benefit for further research and new knowledge.
However, the research idea is good.
In the reviewer’s opinion, the manuscript has important deficiencies that prevent its publication at this stage.
The basic structure of the manuscript is good, but the processing is not thoroughly. Part of the discussion of the results is missing.
The part Introduction is very general and does not state the research goal. The overview of research and the scope of references is also very limited. It is advisable to improve the quality of manuscript processing. Table 1 is on two pages.
The experimental program is interesting but has limited informative value.
It is appropriate to state the specific number of samples as well as the basic mechanical properties of concretes - compressive strength, modulus of elasticity or tensile strength.
The part Conclusion is of a general nature/information.
The manuscript does not clearly present the motivation of the research, the area of new knowledge or the benefits for further research.
I recommend a rework and rewrite manuscript.
Author Response
- The topic can be also considered topical, but the article has/presented limited benefit for further research and new knowledge. However, the research idea is good. In the reviewer’s opinion, the manuscript has important deficiencies that prevent its publication at this stage. The basic structure of the manuscript is good, but the processing is not thoroughly. Part of the discussion of the results is missing.
Response:
Thanks for the reviewer’s positive comments. The conclusions have been rewritten, and some new and constructive contents have been added to strengthen the research target and engineering application. The detailed information can be easily found in the revised paper.
- The part Introduction is very general and does not state the research goal. The overview of research and the scope of references is also very limited. It is advisable to improve the quality of manuscript processing. Table 1 is on two pages.
Response:
Thanks for the reviewer’s suggestion. Extra six references has been added, and extra information has been added to enrich the research background and research objective. Table 1 is on one page in the revised manuscript. The detailed information can be found in the revised paper.
- The experimental program is interesting but has limited informative value.
Response:
Some contents have been complemented to enrich the experimental program, including test specification, specimens’ photo, and testing device, etc., which can be found in the revised paper.
- It is appropriate to state the specific number of samples as well as the basic mechanical properties of concretes - compressive strength, modulus of elasticity or tensile strength.
Response:
It has been listed in lines 95 to 96 of the original manuscript (Three specimens of the same type were fabricated, and the mean value of the three data was regarded as a representative value.).
- The part Conclusion is of a general nature/information.
Response:
The conclusions have been rewritten, and some new contents have been added, which can be found in the revised paper.
- The manuscript does not clearly present the motivation of the research, the area of new knowledge or the benefits for further research. I recommend a rework and rewrite manuscript.
Response:
Some new contents have been added to highlight the research motivation in the section of Introduction, which can be found in the revised paper.
For the convenience of the reviewer, all the changes and the supplement in the revised manuscript have been set in highlight.
Reviewer 2 Report
Generally, it is an interesting paper, but the obtained data might be expected and gave no novelty for the studied topic. If the manuscript should be accepted, the detailed description of the motivation of the conducted research and its novelty must be clearly stated. Also following comments should be considered and addressed in a revised manuscript.
How were the concrete mix designed? What was the purpose of concretes’ design? Please, comment and complete in manuscript.
The commercial repellent additives were originally designed for different purposes? If they were designed for improvement of water absorption of concrete I don’t see any reason, why they were tested.
For the moisture content by mass, symbol w is usually used. Symbol ρ is applied in concrete research for density. Please, correct.
Usually, NaCl solution is used in salt corrosion resistance test.
The freezing temperature is usually -20°C or -15°C. It is not clear, how the freezing temperature was chosen.
Are 25 freezing/thawing cycles sufficient for evaluation of concrete resistance and effectiveness of applied water repellents?
It is not clear, how the dynamic elastic modulus was measured.
In the measurement of Ed, the dry samples must be used and moisture presence negatively affects the measured data. Please, comment and described in detail the measurement procedures.
The water absorption coefficient might be calculated based on measured data. Please, see and refer following paper and add the data on the water absorption coefficient.
Feng, A.S. Guimarães, N. Ramos, L. Sun, D. Gawin, P. Konca, C. Hall, J. Zhao, H. Hirsch, J. Grunewald, M. Fredriksson, K.K. Hansen, Z. Pavlík, A. Hamilton, H. Janssen: Hygric properties of porous building materials (VI): a round robin campaign J Building Environment. 185, 107242 (2020).
See line 157 - With the extension of soaking time, the water absorption ratios of each concrete specimen stabilized. Of course, the samples were capillary saturated with water.
The characterization of tested concrete is incomplete. At least information on bulk density and porosity should be given both for reference state and for samples exposed to salt attack.
The measurement of salt solution absorption should be also done.
Author Response
- How were the concrete mix designed? What was the purpose of concretes’ design? Please, comment and complete in manuscript.
Response:
The concrete mixture proportions were designed according to China’s specification JGJ 55-2011 [26], and the three water/cement ratios of 0.4, 0.5, and 0.6 adopted can represent the concrete types in current civil engineering projects. The above information has been replenished in the revised paper.
- The commercial repellent additives were originally designed for different purposes? If they were designed for improvement of water absorption of concrete I don’t see any reason, why they were tested.
Response:
It’s true that the ordinary commercial repellent agents were usually designed just for waterproofing purpose, and not for anti-frozen purpose. However, many scholars and engineers have attempted to improve concrete durability performance (including chloride resistance, carbonation resistance other than water impermeability) by using waterproofing materials, which can be found in the reference of [13-16]. Moreover, some researches have reported that hydrophobic treatments can improve the frost resistance of concrete [17-20]. In this study, the authors want to make a deep investigation whether a simple hydrophobic treatment of ordinary repellent agents on concrete can effectively improve the salt frost resistance of concrete, and some valuable findings were obtained.
- For the moisture content by mass, symbol w is usually used. Symbol ρ is applied in concrete research for density. Please, correct.
Response:
Thanks for the reviewer’s suggestion. The symbol of water absorption ratio “ρ” has been corrected as “wc”.
- Usually, NaCl solution is used in salt corrosion resistance test.
Response:
It’s true that NaCl solution is usually adopted in the salt scaling resistance experiments due to that NaCl was once the most popular deicing agent for pavement in winter. However, with the gradual realization of the corrosion induced by NaCl on reinforced concrete structures, MgCl2 is now gradually adopted by field engineers as a substitute deicing agent of NaCl in current road engineering [7,29] even though MgCl2 may has similar corrosion risk with NaCl on concrete structures. Therefore, to be consistent with the engineering practice as far as possible, MgCl2 was selected as the corrosion medium in this paper.
- The freezing temperature is usually -20°C or -15°C. It is not clear, how the freezing temperature was chosen.
Response:
It has been clearly stated in lines 114 to 115 in the original manuscript (The central temperature of a specimen was controlled between −17 °C and +8 °C during the freezing and thawing processes.), which is according to China’s standard [28].
- Are 25 freezing/thawing cycles sufficient for evaluation of concrete resistance and effectiveness of applied water repellents?
Response:
It’s true that 25 freezing/thawing cycles is not sufficient for evaluation of concrete resistance and effectiveness of applied water repellents. According to GB/T 50082-2009, we just made an inspection and testing at every 25 cycles. The total freezing/thawing cycles will be terminated until the deterioration criterion of concrete was reached, which was listed as shown in lines 124 to 126 (According to the specification in GB/T 50082-2009 [25], salt frost testing can be considered finished when the mass loss ratio of a concrete specimen reaches 5% or its RDME decreases to 60%.).
- It is not clear, how the dynamic elastic modulus was measured.
Response:
It is a conventional method to determine the relative dynamic modulus of elasticity (RDME) of concrete by using ultrasonic sound velocity, and it has been introduced in detail in many literatures [25,28,29]. In order to avoid unnecessary repetition, it is not introduced in detail in the paper. However, necessary reference citation have been added, from which the detailed information can be found.
- In the measurement of Ed, the dry samples must be used and moisture presence negatively affects the measured data. Please, comment and described in detail the measurement procedures.
Response:
During the salt frost tests, the specimens were always kept saturation condition. According to the China’s standard of GB/T 50082-2009 [25], the RDME of concrete was determined only by the ultrasonic sound velocities of concrete before and after saline freeze/thaw cycles. The parameters of specimens’ moisture contents are not used in the paper.
- The water absorption coefficient might be calculated based on measured data. Please, see and refer following paper and add the data on the water absorption coefficient. Feng, A.S. Guimarães, N. Ramos, L. Sun, D. Gawin, P. Konca, C. Hall, J. Zhao, H. Hirsch, J. Grunewald, M. Fredriksson, K.K. Hansen, Z. Pavlík, A. Hamilton, H. Janssen: Hygric properties of porous building materials (VI): a round robin campaign J Building Environment. 185, 107242 (2020).
Response:
The water absorption coefficient is an effective parameter which can reflect the capillary water absorption of a material. However, compared with the water absorption coefficient, the water absorption ratio is simper, which can represent the hydrophobicity of concrete with repellent agents appropriately. Although it’s easy to transfer the water absorption ratio into the water absorption coefficient, the authors don’t think it’s necessary to make such a change. However, the reference suggested is valuable, and it has been cited as a reference.
- See line 157 - With the extension of soaking time, the water absorption ratios of each concrete specimen stabilized. Of course, the samples were capillary saturated with water.
Response:
Yes, as pointed by the reviewer, the water absorption ratios of each concrete stabilized with the extension of soaking time due to capillary saturation with water. However, the corresponding water absorption ratios of concrete with different treatments are different, which reflects the efficacies of different hydrophobic agents.
- The characterization of tested concrete is incomplete. At least information on bulk density and porosity should be given both for reference state and for samples exposed to salt attack. The measurement of salt solution absorption should be also done.
Response:
The reviewer’s suggestion is good. However, according to China’s standard of GB/T 50082-2009 [25], the bulk density and porosity of concrete are not the regular necessary information for evaluation of concrete frost resistance, and due to the limited time for revision, such information cannot be added in the revised manuscript. Similarly, the salt solution absorption of concrete cannot be provided either in the revised manuscript.
For the convenience of the reviewer, all the changes and the supplement in the revised manuscript have been set in highlight.
Reviewer 3 Report
Manuscript deals with hydrophobic treatment of concrete, using silane and organo-silicone emulsion. Similar articles on this topic (hydrophobic treatment of concrete) have been published many times in the past - however, I very much appreciate that this manuscript refers to these previous studies very often and compares itself with them. Therefore, the article can be rated as interesting.
I have the following comments:
1) Technical terms used in the manuscript must be improved (the article generally needs linguistic proofreading). Expressions such as “ordinary Portland cement” are not professional.
2) The first 2 graphs in Figure 7 show that “internal damage” of tested concrete is almost the same for the reference concrete as for the treated concrete - does this mean that the tested treatments have no effect? This would probably not be entirely consistent with previous studies. It is necessary to better explain (the current explanation in the text is insufficient).
3) Chapter 3.6 deserves more expansion - I recommend supplementing, for example, with photos from my own research proving the conclusions stated in the chapter. I see a problem in the fact that the article is relatively short - an extension of Chapter 3.6 would remedy this.
After incorporating the above comments, I recommend the manuscript for publication.
Reviewer 4 Report
Memorandum
Subject: Review, November 7, 2020
Materials
Title: Effect of hydrophobic treatments on improving the 2 salt frost resistance of concrete
Guo Li1 , Chunhua Fan1 , Yajun Lv2*, Fujun Fan2
1 Jiangsu Key Laboratory of Environmental Impact and Structural Safety in Engineering, China University of Mining and Technology, Xuzhou 221116, China; guoli@cumt.edu.cn; ts18030126p31@cumt.edu.cn
2 School of Architecture, North China University of Water Resources and Electric Power, Zhengzhou 450045,
China; Letmealone2020@126.com
*Correspondence: darkdanking@126.com
Comments:
- The authors must include a nomenclature to define parameters and abbreviations used throughout the paper.
- The authors should add a photo of the specimen along with the dimensions should be included.
- Figure 4 shows photos of concrete specimen after salt frost without elaborating on what is being shown, labels are needed on these figures to identify key points of interest.
- Equally as above, figure 5 needs labels to identify what is being observed.
- The authors should have a descriptive schematic illustrating the concrete contact angle and its value, variation, and association with the specimen.
- The conclusion can be improved. In its present state, it resembles one statement without much description of what was accomplished. The authors should opt to using a bullet type statement to list accomplishments made and issues that may have contributed to the results if any exists.
The paper referenced above I sin good format, upon addressing the above minor issues, it should be ready for publication. Additionally, a minor editorial workup and spelling check is recommended.
Author Response
- The authors must include a nomenclature to define parameters and abbreviations used throughout the paper.
Response:
The parameters and abbreviations used in the paper are just a few, and they have been fully introduced or defined in the paper. Thus, the authors don’t think it’s necessary to add an extra nomenclature.
- The authors should add a photo of the specimen along with the dimensions should be included.
Response:
A new figure (Fig.1 in the revised paper) of prism specimens with BS4004 has been added. Because the two types of specimens (100´100´100 mm3 and 100´100´400 mm3) adopted in the paper are just very common shapes, the authors don’t think it’s necessary to add all the photos of specimens.
Fig. 1 Prism concrete specimens applied with a repellent agent of BS4004
- Figure 4 shows photos of concrete specimen after salt frost without elaborating on what is being shown, labels are needed on these figures to identify key points of interest.
Response:
Labels have been added in the revised Fig. 4.
- Equally as above, figure 5 needs labels to identify what is being observed.
Response:
Labels have been added in the revised Fig. 5.
- The authors should have a descriptive schematic illustrating the concrete contact angle and its value, variation, and association with the specimen.
Response:
Some descriptive contents have been added in the revised paper [27,30].
- The conclusion can be improved. In its present state, it resembles one statement without much description of what was accomplished. The authors should opt to using a bullet type statement to list accomplishments made and issues that may have contributed to the results if any exists.
Response:
Thanks for the reviewer’s valuable suggestion. The conclusions have been rewritten, and some new and constructive findings has been added. The detailed information can be found in the revised paper.
- The paper referenced above I sin good format, upon addressing the above minor issues, it should be ready for publication. Additionally, a minor editorial workup and spelling check is recommended.
Response:
Thanks for the reviewer’s positive comments. A thorough check including spelling and grammar has been made on the revised manuscript.
For the convenience of the reviewer, all the changes and the supplement in the revised manuscript have been set in highlight.
Reviewer 5 Report
1) The originality and the scientific value of the subject are good. Indeed, an important problem having direct applications is treated.
2) The Abstract is concrete as it gives the summary of this research work in a concise manner. In addition, it is sufficiently supported by the results obtained during research.
3) The Introduction Section in its current form is not adequate. In this context, I recommend the authors to further analyze and discuss the results of Refs. [1-3], [4-10], [11,12] and [13-15]. Besides, the differences/advantages of the present investigation compared to other literature works should be written out at the end of this Section in a much more detailed and comprehensive manner.
4) The materials, their applications, applied methods and especially the use of the investigated material are explained in detail. The composition, the origin of the material used, dimensions of specimens etc are all mentioned.
5) Presentation of the experimental work is very thorough. Process and prerequisites of sample preparation are clearly mentioned. However, the authors are kindly recommended to provide some further technical details about the laboratory equipment that they used to carry out their experiments.
6) The performance and clarity of results and data are good. Yet, the discussion of the results is relatively adequate. The authors could give some additional theoretical explanations about Figs. 2 (a,b,c), 3 (a,b,c) and 4(a,b,c,d,e,f,gh,i).
In addition, the quality of Fig. 4a is substandard and evidently not in publication level.
The authors are kindly recommended to address this issue.
Moreover, is there any possibility for comparison with advanced computational methods (FEM, BEM)?
7) Logic and coherence are concrete and the clarity and quality of writing are sound.
8) The Conclusions Section performs the findings of this work in a rather brief manner. It should become more thorough and detailed.
Also, I invite the authors to add a paragraph on the motives and prospects that this work provides for future research.
Overall, it is the reviewer’s opinion that this paper may be recommended for publication provided that the authors interpret these critical remarks in a constructive manner and revise the manuscript accordingly.
Author Response
- 1.The originality and the scientific value of the subject are good. Indeed, an important problem having direct applications is treated.
Response:
Thanks for the reviewer’s positive comments.
- 2.The Abstract is concrete as it gives the summary of this research work in a concise manner. In addition, it is sufficiently supported by the results obtained during research.
Response:
Thanks for the reviewer’s positive comments.
- 3.The Introduction Section in its current form is not adequate. In this context, I recommend the authors to further analyze and discuss the results of Refs. [1-3], [4-10], [11,12] and [13-15]. Besides, the differences/advantages of the present investigation compared to other literature works should be written out at the end of this Section in a much more detailed and comprehensive manner.
Response:
Thanks for the reviewer’s suggestion. Six new references have been replenished, and some new discussion have been added to highlight the research background and research target in the section of Introduction, which can be found in the revised manuscript.
- 4.The materials, their applications, applied methods and especially the use of the investigated material are explained in detail. The composition, the origin of the material used, dimensions of specimens etc are all mentioned.
Response:
Thanks for the reviewer’s positive comments.
- 5.Presentation of the experimental work is very thorough. Process and prerequisites of sample preparation are clearly mentioned. However, the authors are kindly recommended to provide some further technical details about the laboratory equipment that they used to carry out their experiments.
Response:
Thanks for the reviewer’s suggestion, some information about the rapid freezing/thawing cycle machine used has been added. The device adopted for the concrete salt frost experiments was a TR-TDRF-1 rapid freeze-thaw cycle machine produced by Shanghai Torrent Instrument Co. LTD.
- 6.The performance and clarity of results and data are good. Yet, the discussion of the results is relatively adequate. The authors could give some additional theoretical explanations about Figs. 2 (a,b,c), 3 (a,b,c) and 4(a,b,c,d,e,f,gh,i).
In addition, the quality of Fig. 4a is substandard and evidently not in publication level.
The authors are kindly recommended to address this issue.
Moreover, is there any possibility for comparison with advanced computational methods (FEM, BEM)?
Response:
Thanks for the reviewer’s positive evaluation and kind suggestion. Some additional theoretical explanations and discussion for Figs. 2, 3 and 4 have been added, or there have been already existed, which can be found in the revised manuscript.
The quality of Fig. 4a and Figs 4b, 4c, 4d,4e,4f,4g, 4h, and 4i are the same, the authors don’t think it is substandard for publication. Because Fig. 4a is the photo of untreated concrete before salt frost test, it’s slightly different with the appearance of concrete applied with repellent agents (Figs. 4d and 4g ).
As about FEM and BEM analysis, it’s a good suggestion. However, due to the limited time for revision, the FEM or BEM analysis can’t be accomplished now. Maybe in the future studies, it can be conducted.
- 7.Logic and coherence are concrete and the clarity and quality of writing are sound.
Response:
Thanks for the reviewer’s positive comments.
- 8.The Conclusions Section performs the findings of this work in a rather brief manner. It should become more thorough and detailed.
Also, I invite the authors to add a paragraph on the motives and prospects that this work provides for future research.
Overall, it is the reviewer’s opinion that this paper may be recommended for publication provided that the authors interpret these critical remarks in a constructive manner and revise the manuscript accordingly.
Response:
Thanks for the reviewer’s valuable suggestion. Some new and constructive contents have been added in the conclusions, and the conclusion part has almost been rewritten.
As about the prospects for the future research, some extra contents have also been replenished (Additionally, with the emergence of novel superhydrophobic materials, such superhydrophobic treatment can greatly reduce the water absorption of concrete [36-37], which may produce satisfactory results on the salt frost resistance of concrete, and needs further studies in the future.).
For the convenience of the reviewer, all the changes and the supplement in the revised manuscript have been set in highlight.
Round 2
Reviewer 1 Report
Thanks for the answers and manuscript improvements.
The manuscript and research are in their current form better understandable and interesting for potential readers.
However, during the review, I found shortcomings.
Figure 1. is very small.
Figure 3. has a label on the next page.
Figure 7. is on two pages.
Figure 8. is on two pages.
from the last Review:
point 4. It is appropriate to state the specific number of samples as well as the basic mechanical properties of concretes - compressive strength, modulus of elasticity or tensile strength.
This point is not sufficiently resolved.
I recommend to state measured values, mean, standard deviation, etc.
Overall, it is also possible to improve the presentation of results.
After editing, it will be possible to publish the article.
Author Response
Comments and response of reviewer 1(2)
Thanks for the answers and manuscript improvements. The manuscript and research are in their current form better understandable and interesting for potential readers. However, during the review, I found shortcomings. Figure 1. is very small. Figure 3. has a label on the next page. Figure 7. is on two pages. Figure 8. is on two pages.
Response:
Thanks for the reviewer’s suggestion. Fig. 1 has been enlarged a little. As for the layout problems of Figs. 3, 7 and 8, the authors think that it’s not the task of the authors, the corresponding editor will deal with it appropriately before publication.
From the last Review:
point 4. It is appropriate to state the specific number of samples as well as the basic mechanical properties of concretes - compressive strength, modulus of elasticity or tensile strength. This point is not sufficiently resolved. I recommend to state measured values, mean, standard deviation, etc. Overall, it is also possible to improve the presentation of results.
After editing, it will be possible to publish the article.
Response:
Thanks for the reviewer’s suggestion. As for specimens’ specific number, it has been revised as below and in the revised manuscript (line 113).
Three specimens of the same type in one test were fabricated, and the mean value of the three data was regarded as a representative value.
As for concrete compressive strength, elastic modulus and tensile strength, it’s a good idea to complement them. However, according to the China’s standard of GB/T 50082-2009, the evaluation parameters for the damages of concrete in a rapid freeze-thaw cycle test are the mass loss ratio and relative dynamic modulus of elasticity of concrete, thus the basic mechanical properties of concrete including compressive strength, elastic modulus and tensile strength were not tested. As a result, it can’t be provided in the revised manuscript.
Reviewer 2 Report
As most of my comments and sugestions were considered in revised paper, I recommend its publication in present form.
Author Response
As most of my comments and suggestions were considered in revised paper, I recommend its publication in present form.
Response:
Thanks.
Reviewer 3 Report
Comments from the first round of reviews were incorporated, only one partial deficiency remained (one word). The author still uses the term "ordinary Portland cement" in the manuscript and ignored my comment on it. None of the cements can be marked as "ordinary". There are many types of cements and in the professional article it is necessary to specify the type of used cement. For example international standard EN 197-1 distinguishes 5 basic types of cements (CEM I - CEM V) and many subclasses. Please specify the exact type of used cement (not "ordinary Portland cement").
After incorporating this last detail, I recommend the manuscript for publication.
Author Response
Comments and response of reviewer 3(2)
Comments from the first round of reviews were incorporated, only one partial deficiency remained (one word). The author still uses the term "ordinary Portland cement" in the manuscript and ignored my comment on it. None of the cements can be marked as "ordinary". There are many types of cements and in the professional article it is necessary to specify the type of used cement. For example international standard EN 197-1 distinguishes 5 basic types of cements (CEM I - CEM V) and many subclasses. Please specify the exact type of used cement (not "ordinary Portland cement").
After incorporating this last detail, I recommend the manuscript for publication.
Response:
Thanks for the reviewer’s suggestion, and some revision has been made as shown below. There is some misunderstanding about the name of the cement adopted in this study. “Ordinary Portland cement” is the name of a Chinese cement type according to China’s standard GB 175-2007 (Common Portland Cement), and such cement is similar to the cement type of CEM II/A-M according to EN 197-1.
“Among the concrete raw materials, the cement was P·O 42.5 ordinary Portland cement according to China’s specification GB 175-2007 [27].”
Reviewer 5 Report
I read the revised manuscript very carefully and came to the conclusion that the authors made a significant endeavor to answer all critical remarks and improve their paper effectively.
I believe that the article in its current form is in compliance with the high quality standards of “Materials”.
Author Response
Comments and response of reviewer 5(2)
I read the revised manuscript very carefully and came to the conclusion that the authors made a significant endeavor to answer all critical remarks and improve their paper effectively.
I believe that the article in its current form is in compliance with the high quality standards of “Materials”.
Response:
Thanks.